# A comparative analysis of APGAR score and the gold standard in the diagnosis of birth asphyxia at a tertiary health facility in Kenya

Albertine Enjema Njie[1]*, Winstone Mokaya Nyandiko[1,2], Phinehas Ademi Ahoya[3], Jude Suh Moutchia[4]

1 Department of Child Health and Pediatrics, College of Health Sciences—Moi University, Eldoret, Kenya, 2 Academic Model Providing Access to Healthcare, Eldoret, Kenya, 3 Directorate of Child Health and Pediatrics, Moi Teaching and Referral Hospital, Eldoret, Kenya, 4 Department of Biostatistics, Epidemiology and Informatics -Perelman School of Medicine at the University of Pennsylvania, Philadelphia, Pennsylvania, United States of America

* njieenjema@gmail.com

## Abstract

### Background

Birth asphyxia is a consistent key contributor to neonatal morbidity and mortality, notably in sub-Saharan Africa. The APGAR score, though a globally used diagnostic tool for birth asphyxia, remains largely understudied especially in resource-poor settings.

### Objective

This study determined how effectively the APGAR score is used to diagnose birth asphyxia in comparison to the gold standard (umbilical cord blood pH <7 with neurologic involvement) at Moi Teaching and Referral Hospital (MTRH), and identified healthcare provider factors that affect ineffective use of the score.

### Methods

Using a quantitative cross-sectional hospital-based design, term babies born in MTRH who weighed ≥2500g were randomly and systematically sampled; and healthcare providers who assign APGAR scores were enrolled via a census. Umbilical cord blood was drawn at birth and at 5minutes for pH analysis. APGAR scores assigned by healthcare providers were recorded. Effective use of the APGAR score was determined by sensitivity, specificity, positive and negative predictive values. At a significance level of 0.05, multiple logistic regression analysis identified the independent provider-associated factors affecting ineffective use of the APGAR score.

### Results

We enrolled 102 babies, and 50 (49%) were females. Among the 64 healthcare providers recruited, 40 (63%) were female and the median age was 34.5years [IQR: 31.0, 37.0]. Assigned APGAR scores had a sensitivity of 71% and specificity of 89%, with positive and

**Funding:** The author(s) received no specific funding for this work.

**Competing interests:** The authors have declared that no competing interests exist.

negative predictive values of 62% and 92% respectively. Healthcare provider factors associated with ineffective APGAR score use included: instrumental delivery (OR: 8.83 [95% CI: 0.79, 199]), lack of access to APGAR scoring charts (OR: 56.0 [95% CI: 12.9, 322.3]), and neonatal resuscitation (OR: 23.83 [95% CI: 6.72, 101.99]).

## Conclusion

Assigned APGAR scores had low sensitivity and positive predictive values. Healthcare provider factors independently associated with ineffective APGAR scoring include; instrumental delivery, lack of access to APGAR scoring charts, and neonatal resuscitation.

## Introduction

Perinatal mortality is still a significant public health issue, with its top three aetiologies identified as; prematurity (28%), neonatal infections (26%), and birth asphyxia (23%) [1,2]. Birth asphyxia is a neonatal condition characterized by failure to initiate or sustain breathing at birth, and it has been a consistent major aetiology of neonatal deaths in the last decade [3]. Globally, neonatal mortality accounts for about half of under-five mortality [4]. From 2003–2014, childhood mortality trends in Kenya showed that, among neonatal, infant, and under-5 mortalities, neonatal deaths showed the slowest decrease [5]). Therefore, to decrease the mortality rate of children below five years of age, it is imperative to decrease these neonatal deaths [6]. To further curb neonatal mortality, there is a profound need to explore its key aetiologies such as birth asphyxia, especially in resource-poor settings [7]. Sub-Saharan countries bear the greatest burden of birth asphyxia, and these asphyxia-related deaths are for the most part preventable [8].

The APGAR score, though widely used for birth asphyxia diagnosis can either overestimate or underestimate asphyxia due to its subjective nature [9]. The combined APGAR score has shown some superiority to the conventional APGAR score in birth asphyxia diagnosis [10,11], and over the years, severely asphyxiated babies have been found to have normal conventional APGAR scores [9,12]. However, in low- and middle- income countries, the conventional APGAR score continues to be solely used for birth asphyxia diagnosis. An APGAR score of <7 depicts asphyxia; to ascribe this asphyxia to an intrapartum hypoxic-ischemic event, the American College of Obstetricians and Gynecologists and American Academy of Pediatrics require an add-on umbilical cord pH value of <7, evidence of neurologic involvement (e.g., seizures, altered tone, coma) and evidence of multiorgan involvement [13]. Umbilical cord blood pH has also been shown to have better diagnostic value in birth asphyxia [14,15], and when combined with the APGAR score, higher sensitivity in birth asphyxia diagnosis has been observed [16,17].

Birth asphyxia is a major contributor to neonatal morbidity and mortality. Globally, it contributes to a quarter of all neonatal deaths. In Kenya, according to the KDHS 2014 report, birth asphyxia is the leading cause of neonatal deaths at 31.6% [5]. Its consistent unfavorable effect on neonatal mortality in the last decade [18], highlights the need to assess the performance of APGAR score as a diagnostic tool for birth asphyxia in comparison to the gold standard. Failure to optimally diagnose birth asphyxia directly interferes with the timeline of managing these neonates, thereby increasing the risk of neonatal mortality and development of asphyxia-related morbidity.

Published studies report varying proportions of APGAR score sensitivities and specificities of 62.8–99% and 41–81% respectively [11,19,20]. Consistency in APGAR scoring among healthcare providers has been shown to also vary greatly, with the heart rate component of the score having the least inconsistency. Healthcare provider factors influencing APGAR scoring include among others: years of experience of the healthcare provider, cadre of the healthcare provider, number of staff per delivery, nature of resuscitation done by healthcare provider, and the type of delivery [21–23]. Exploring these factors in this study is pivotal in fostering improved usage of the APGAR score by healthcare providers in identifying neonates with asphyxia.

Improvements in antenatal care, delivery and neonatal resuscitation services have been observed in many countries with developing economies. This has led to the establishment of national intervention programs such as "Help Babies Breathe" in Kenya [24,25]. Yet, birth asphyxia consistently remains a major cause of neonatal mortality, creating the need for effective use of its diagnostic tools such as the APGAR score. If the diagnosis of birth asphyxia is not effectively done, it cannot be promptly and adequately managed. Consequently, its short- and long-term morbidity, and mortality persistently occur as reflected by the above-mentioned data. This study therefore aimed to determine how effectively the APGAR score is applied in the labour ward and theatre of a tertiary hospital in Kenya. Furthermore, it assessed healthcare provider factors associated with this effective use of APGAR score. This knowledge will aid in addressing and halting asphyxia-related morbidity and mortality through evidence-based data.

## Materials and methods

This was an analytical cross-sectional hospital-based quantitative study carried out in the labour ward, delivery rooms and theatre of Moi Teaching and Referral Hospital (MTRH), Eldoret, between June 2021 and October 2021. The hospital is a level 6 public hospital and is currently the second largest national teaching and referral hospital in Kenya. Its delivery unit, obstetrics theatre and newborn unit (NBU) are located at the Riley Mother and Baby wing of the hospital. An average of 750 deliveries are carried out and approximately 250 babies are admitted to NBU monthly, creating an optimal study setting for birth asphyxia.

The Karimollah formula for accuracy studies [26] was used to estimate the sample size for this study, using a pre-study estimate of 81% sensitivity and specificity obtained by Dalili and collaborates in their study [11]. The sampling interval (k) calculated for this study was 2. Therefore, the researchers randomly and systematically sampled 102 term neonates (37-42weeks gestation) who had a birthweight of at least 2500 grams. Neonates with life-threatening malformations were excluded because the APGAR score has been shown to be influenced by congenital malformations [15,27]; while, neonates at risk of anaemia/sepsis in the immediate postnatal period were excluded as these could alter umbilical cord blood pH values [28]. Details on neonatal recruitment are demonstrated on Fig 1 below.

To assess healthcare provider factors, 64 healthcare providers who assign APGAR scores to neonates after delivery were enrolled via a census, due to the definite number of healthcare providers in the unit. Interviewer-administered semi-structured questionnaires (*S1 File*) were used to collect neonatal sociodemographic and clinical characteristics. To collect information on healthcare provider sociodemographic and factors influencing APGAR scoring among healthcare workers, a self-administered questionnaire (*S2 File*) was used. Umbilical cord blood was collected at birth and in the fifth minute of life from the neonates for pH analysis. This was in accordance to the Standard operating procedure (SOP) adapted from the 2015 guidelines of the Research Centre for Women's and Infants' Health (RCWIH) BioBank in Canada for umbilical cord blood collection (*S3 File*). APGAR scores in the first, fifth and tenth minutes

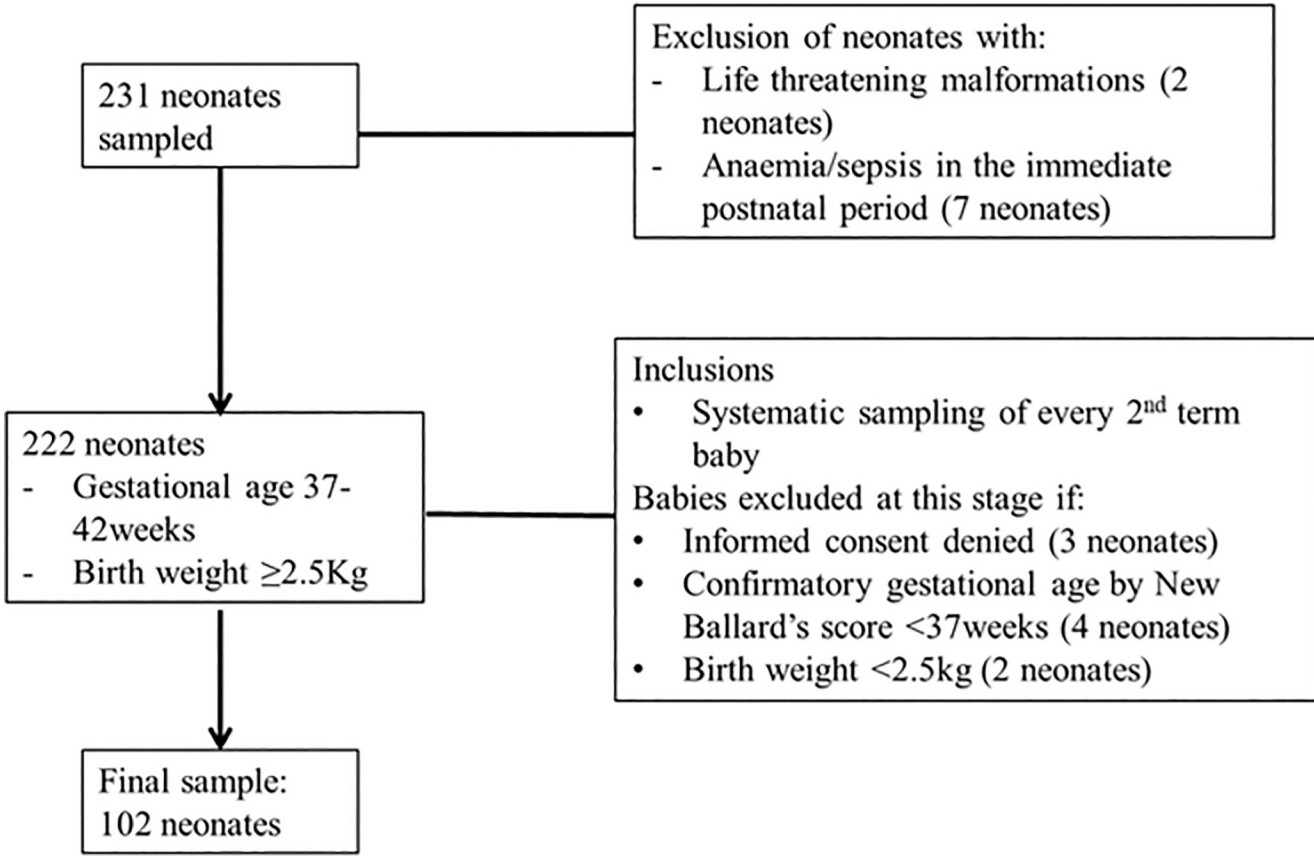

**Fig 1. Neonatal inclusion flow chart.**

as routinely given after delivery by healthcare providers were documented; as well as data on neonates who had altered tone and/or clinically apparent seizures within the first 24 hours of life.

Counts and proportions were used to summarise categorical data while median and inter-quartile range were used to summarise continuous variables. The chi squared test (or Fisher's exact test if any expected cell count was <5) was used to assess significant differences in categorical variables across asphyxia and no asphyxia groups. The Wilcoxon rank sum test was used to test association between predictors and outcomes for continuous variables. Effective APGAR score use was computed as sensitivity, specificity, positive and negative predictive values. Intraclass correlations did not identify the need for multi-level regression models in this study; thus, one-level logistic regression was used at a 95% confidence level to determine independent factors associated with ineffective APGAR score use, with the dependent variable being incorrect APGAR scores.

This study received ethical approval from the Institutional Review and Ethics Committee of Moi Teaching and Referral Hospital / Moi University School of Medicine (Approval 0003781). Additionally, an administrative approval was obtained from the Chief Executive Officer (CEO) of Moi Teaching and Referral Hospital (Ref. ELD/MTRH/R&P/10/2/V.2/2010), and a research permit was gotten from the National Commission for Science, Technology, and Innovation Kenya (Ref. 791482). Prior to enrolment of neonates, a written informed consent was obtained from the parents of neonates who met eligibility to participate in this study. Furthermore, healthcare workers provided a written informed consent before partaking in this study.

## Results

This study enrolled 102 neonates, of whom, slightly more than half were male, 52 (51%). The median [IQR] birth weight and gestational age were 3,010g [IQR: 2,752, 3,375] and 39.0 [IQR: 37.0, 40.2] weeks respectively. About two-thirds of the neonates, 65 (63.7%), were born via spontaneous vaginal delivery. Labour was induced in 11 deliveries (10.8%), with the most frequent indication being premature rupture of membranes (5 [45.5%]), seconded by postdate pregnancies (4 [36.4%]). The median [IQR] APGAR score at the 5th minute was 9.0 [IQR: 8.0, 10.0], and 24 (23.5%) neonates were observed to have an APGAR score of <7 at this 5th minute. The median [IQR] cord blood pH at birth was 7.20 [7.09, 7.29] and 7.20 [7.13, 7.28] at the 5th minute. Eleven neonates (10.8%) had clinically apparent seizures, while 17 (16.7%) were noted to have altered tone (Table 1).

Among all the 102 neonates enrolled, 21 (21%) had birth asphyxia defined as cord blood pH at birth <7 and presence of clinically apparent seizures and/or an altered tone. The median [IQR] APGAR score was significantly lower amongst neonates with birth asphyxia compared to neonates without birth asphyxia (*p* <0.001) at birth (7.0 [IQR: 6.0, 7.0] vs. 9.0 [IQR: 8.0,

**Table 1. Neonatal sociodemographic and clinical characteristics.**

| Characteristic | Overall, N = 102 | No asphyxia, N = 81 | Asphyxia[1], N = 21 | p-value[2] |
|---|---|---|---|---|
| **Sex of the neonate, n (%)** | | | | **0.04** |
| Female | 50.0 (49.0) | 44.0 (54.3) | 6.0 (28.6) | |
| Male | 52.0 (51.0) | 37.0 (45.7) | 15.0 (71.4) | |
| **Weight (g), median [IQR]** | 3,010 [2,752, 3,375] | 3,015 [2,795, 3,320] | 3,000 [2,620, 3,500] | 0.82 |
| **Gestational age (months), median [IQR]** | 39.0 [37.0, 40.2] | 39.0 [37.0, 40.1] | 39.5 [38.0, 41.0] | 0.20 |
| **Induction of labour, n (%)** | 11.0 (10.8) | 7.0 (8.6) | 4.0 (19.0) | 0.23 |
| **[1] Indication for induction of labour, n (%)** | | | | 0.27 |
| Post date | 4.0 (36.4) | 3.0 (42.9) | 1.0 (25.0) | |
| PROM | 5.0 (45.5) | 4.0 (57.1) | 1.0 (25.0) | |
| SPET | 2.0 (18.2) | 0.0 (0.0) | 2.0 (50.0) | |
| **Method of delivery, n (%)** | | | | **0.014** |
| Caesarean section (CS) | 34.0 (33.3) | 28.0 (34.6) | 6.0 (28.6) | |
| Instrumental vaginal delivery (IVD) | 3.0 (2.9) | 0.0 (0.0) | 3.0 (14.3) | |
| Normal vaginal delivery | 65.0 (63.7) | 53.0 (65.4) | 12.0 (57.1) | |
| **APGAR score at birth** | | | | |
| Median [IQR] | 8.0 [7.3, 9.0] | 9.0 [8.0, 9.0] | 7.0 [6.0, 7.0] | **<0.001** |
| APGAR score < 7, n (%) | 26.0 (25.5) | 10.0 (12.3) | 16.0 (76.2) | **<0.001** |
| **5th minute APGAR score** | | | | |
| Median [IQR] | 9.0 [8.0, 10.0] | 9.0 [9.0, 10.0] | 7.0 [6.0, 8.0] | **<0.001** |
| APGAR score < 7, n (%) | 24.0 (23.5) | 9.0 (11.1) | 15.0 (71.4) | **<0.001** |
| **10th minute APGAR score** | | | | |
| Median [IQR] | 10.0 [9.0, 10.0] | 10.0 [10.0, 10.0] | 8.0 [8.0, 10.0] | **<0.001** |
| APGAR score < 7, n (%) | 6.0 (5.9) | 2.0 (2.5) | 4.0 (19.0) | **0.016** |
| **Cord blood pH at birth, median [IQR]** | 7.20 [7.09, 7.29] | 7.23 [7.14, 7.32] | 7.00 [6.97, 7.00] | **<0.001** |
| **Cord blood pH at the 5th minute, median [IQR]** | 7.20 [7.13, 7.28] | 7.21 [7.15, 7.30] | 7.12 [7.06, 7.16] | **<0.001** |
| **Clinically apparent seizures, n (%)** | 11.0 (10.8) | 0.0 (0.0) | 11.0 (52.4) | **<0.001** |
| **Altered tone, n (%)** | 17.0 (16.7) | 0.0 (0.0) | 17.0 (81.0) | **<0.001** |

[1] Amongst the 11 babies born post induced labour.

[2] p values from Wilcoxon rank sum test, Fisher's exact test, and Pearson's Chi-squared test as appropriate.

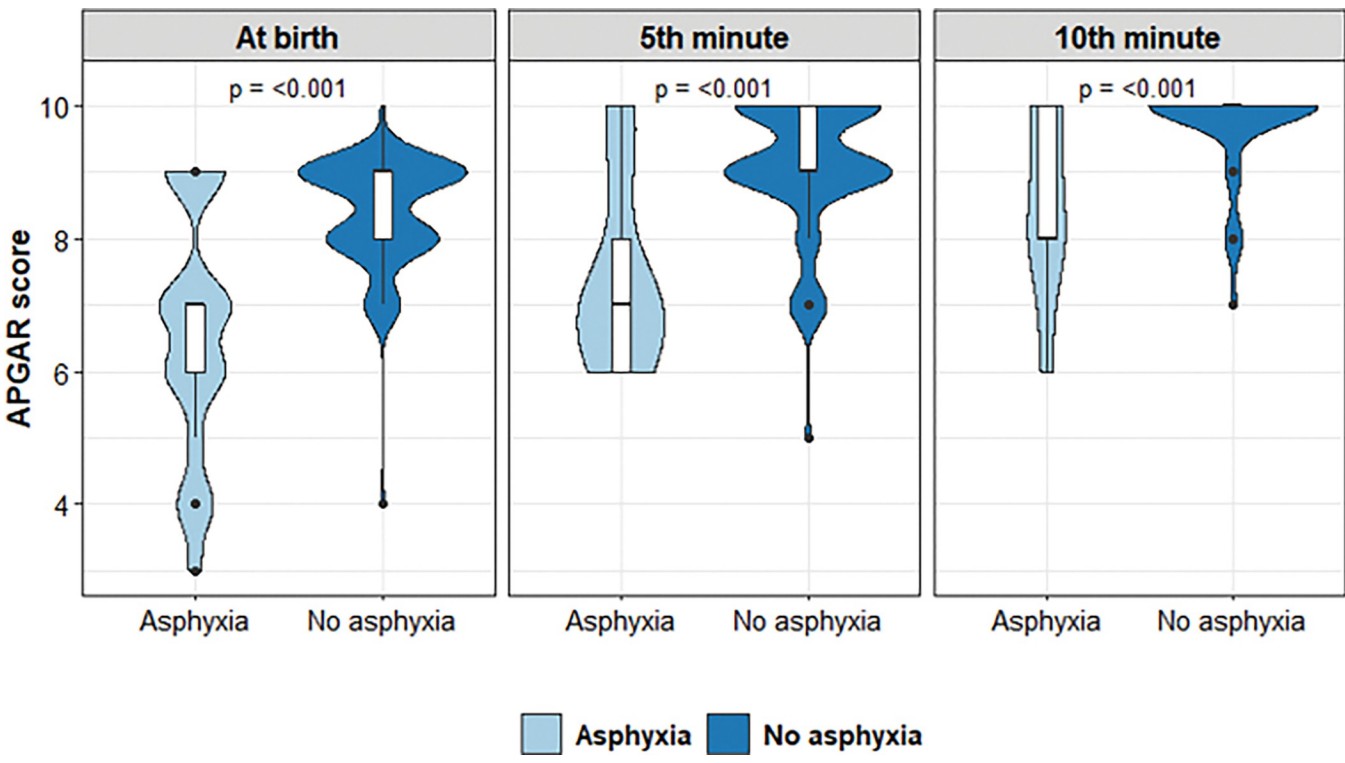

**Fig 2. Violin plots showing the distribution of APGAR score at various time points by birth asphyxia status.**

9.0]), at the 5[th] minute (7.0 [IQR: 6.0, 8.0] vs. 9.0 [IQR: 9.0, 10.0]), and at the 10[th] minute (8.0 [IQR: 8.0, 10.0] vs. 10.0 [IQR: 10.0, 10.0])–Fig 2.

A total of 64 healthcare workers assigned APGAR scores to the 102 neonates. The median [IQR] age of the healthcare providers was 34.5 [31.0, 37.0] years and 40 (63%) were female. Of these healthcare workers, there were 13 (20%) students, 3 (5%) medical officers, 31 (49%) mid-wives, 13 (20%) paediatric residents, and 4 (6%) paediatric consultants. Half (50%) of the healthcare providers had between five- and ten-years working experience with the minimum (2%) observed to have worked for more than 15 years (Table 2).

Healthcare providers agreed to considering the following when assigning APGAR scores: 59 (92%) consider the individual parameters of the score, 45 (70%) seek a second opinion before assigning scores and 47 (73%) routinely consider the second opinions, 57 (89%) consider the need for resuscitation, 54 (84%) consider the nature of the resuscitation, 46 (72%) consider the duration of the resuscitation, 32 (50%) consider the time of the day, and 42 (66%) typically consider the type of delivery. Fifty-one (80%) reported having ready access to the APGAR scoring chart; however, only 24 (38%) felt the need to refer to the APGAR scoring chart before assigning a score to a neonate.

Table 3A is a 2 x 2 table of birth asphyxia as diagnosed by an assigned APGAR score of <7 at the 5[th] minute versus birth asphyxia as diagnosed by a cord blood pH of <7 at birth with the presence of clinically apparent seizures and/or an altered tone (gold standard).

Of the neonates with birth asphyxia, 15 were truly identified by APGAR scoring at the 5[th] as having birth asphyxia (sensitivity: 0.71 [95% CI: 0.48, 0.89]), and of the neonates without birth asphyxia, 72 were truly identified by APGAR scoring at the 5[th] minute as not having birth asphyxia (specificity: 0.89 [95% CI: 0.80, 0.95]) as shown on Table 3B.

**Table 2. Socio-demographic characteristics of the healthcare providers who scored the neonates.**

| Characteristic | Total, N = 64 |
| --- | --- |
| **Age (years), median [IQR]** | 34.5 [31.0, 37.0] |
| **Sex, n (%)** | |
| Female | 40.0 (62.5%) |
| Male | 24.0 (37.5%) |
| **Cadre, n (%)** | |
| Medical officer | 3.0 (4.7%) |
| Midwife | 31.0 (48.5%) |
| Paediatric consultant | 4.0 (6.2%) |
| Paediatric resident | 13.0 (20.3%) |
| Student | 13.0 (20.3%) |
| [1]**Education, n (%)** | |
| Diploma | 14.0 (45.2%) |
| Higher National Diploma | 4.0 (12.9%) |
| Bachelor's degree | 12.0 (38.7%) |
| Master's degree | 1.0 (3.2%) |
| **Years of work experience, n (%)** | |
| <5 years | 20.0 (31.3%) |
| 5–10 years | 32.0 (50.0%) |
| >10–15 years | 11.0 (17.2%) |
| >15 years | 1.0 (1.6%) |

[1]Amongst midwives.

Neonates born via instrumental vaginal delivery had 8.83 times higher odds (95% CI: 0.79, 199) of having an incorrect asphyxia classification compared to neonates born via normal vaginal delivery. In resuscitated neonates, there was a 23.83 (6.72, 101.99) times significantly higher odds of an incorrect asphyxia classification; however, the nature of resuscitation done was not a statistically significant finding. Healthcare practitioners without access to an APGAR scoring chart were significantly more likely (OR: 56.0 [95% CI: 12.9, 322.3]) to incorrectly classify

**Table 3. a: Diagnosis of birth asphyxia using APGAR score at the 5th minute versus cord pH at birth.** b: APGAR scoring at the 5[th] minute for the diagnosis of birth asphyxia.

| | | Cord pH at birth with clinically apparent seizures and/or altered tone | | |
| --- | --- | --- | --- | --- |
| | | Asphyxia | No asphyxia | Total |
| **APGAR score at the 5th minute** | Asphyxia | 15 (**True Positives**) | 9 (False Positives) | 24 |
| | No asphyxia | 6 (False Negatives) | 72 (**True Negatives**) | 78 |
| | **Total** | 21 | 81 | **102** |

| Parameter | Value (95% CI) |
| --- | --- |
| **Accuracy** | 0.85 (0.77, 0.92) |
| **Sensitivity** | 0.71 (0.48, 0.89) |
| **Specificity** | 0.89 (0.80, 0.95) |
| **Positive predictive value** | 0.62 (0.41, 0.81) |
| **Negative predictive value** | 0.92 (0.84, 0.97) |
| **Positive likelihood ratio** | 6.43 (3.28, 12.60) |
| **Negative likelihood ratio** | 0.32 (0.16, 0.63) |

**Table 4. Factors associated with an incorrect asphyxia classification based on APGAR scoring.**

| Variable | OR (95% CI) | p value | [1] aOR (95% CI) | p value |
|---|---|---|---|---|
| **Neonate related factors** | | | | |
| Sex (Female) | 0.89 (0.29, 2.70) | 0.84 | 0.85 (0.27, 2.62) | 0.78 |
| Birth weight | 0.99 (0.99, 1.00) | 0.73 | 1.00 (0.99, 1.00) | 0.88 |
| **Healthcare practitioner related factors** | | | | |
| Age (in years) | 1.06 (0.95, 1.20) | 0.35 | 1.06 (0.95, 1.20) | 0.32 |
| Sex (Female) | 0.60 (0.20, 1.87) | 0.38 | 0.86 (0.19, 1.81) | 0.34 |
| **Cadre** | | 0.74 | | 0.35 |
| Student | Ref. | - | Ref. | - |
| Midwife | 1.75 (0.38, 12.41) | 0.51 | 2.34 (0.49, 17.08) | 0.33 |
| Medical officer | 2.75 (0.11, 36.59) | 0.45 | 1.05 (0.34, 223.03) | 0.13 |
| Paediatric resident | 3.38 (0.58, 26.96) | 0.19 | 6.87 (0.98, 65.03) | 0.06 |
| Paediatric consultant | 2.75 (0.11, 36.59) | 0.45 | 7.09 (0.25, 124.4) | 0.18 |
| **Education level (midwives only)** | | 0.45 | | 0.49 |
| Diploma | Ref. | - | Ref. | - |
| Higher National Diploma | 0.51 (0.16, 1.72) | 0.26 | 0.53 (0.16, 1.83) | 0.30 |
| Bachelor's degree | 0.19 (0.01, 1.24) | 0.14 | 0.19 (0.01, 1.34) | 0.15 |
| Master's degree | - | - | - | - |
| **Years of experience** | | 0.75 | | 0.77 |
| <5 years | Ref. | - | Ref. | - |
| 5–10 years | 0.29 (0.01, 1.92) | 0.27 | 0.31 (0.02, 2.2) | 0.31 |
| >10–15 years | - | - | | |
| >15 years | 0.89 (0.28, 2.96) | 0.84 | 0.99 (0.30, 3.43) | 0.98 |
| **Method of delivery** | | 0.03 | | **0.04** |
| Normal vaginal delivery | Ref. | - | Ref. | - |
| Caesarean section (CS) | 0.13 (0.01, 0.73) | 0.05 | 0.12 (0.01, 0.77) | 0.06 |
| Instrumental vaginal delivery (IVD) | 8.83 (0.79, 199) | 0.08 | 9.87 (0.81, 236) | 0.08 |
| **Individual APGAR score parameters** | 0.41 (0.02, 2.31) | 0.27 | 0.42 (0.02, 2.45) | 0.43 |
| **The need for resuscitation** | 23.83 (6.72, 101.99) | <0.001 | 27.82 (7.29, 134.80) | **<0.001** |
| **Nature of resuscitation** | | 0.99 | | 0.98 |
| Suctioning/stimulation | Ref. | - | Ref. | - |
| Bag/mask ventilation | 1.00 (0.07, 13.85) | 1.00 | 0.67 (0.01, 12.4) | 0.78 |
| Chest compressions | 0.00 (-, -) | 0.99 | | 0.99 |
| Intubation | 0.00 (-, -) | 0.99 | | 0.99 |
| Oxygenation | 0.67 (0.04, 10.03) | 0.77 | 0.35 (0.01, 7.22) | 0.52 |
| Use of adrenaline | 0.00 (-, -) | 0.99 | | 0.99 |
| **Lack of access to APGAR score chart** | 56.0 (12.9, 322.3) | <0.001 | 78.9 (15.5, 661.7) | **<0.001** |

[1]Odds ratio adjusted for gestational age, sex of the neonate, and the birth weight of the neonate, one-level logistic regression analysis.

asphyxia compared to healthcare practitioners with access to an APGAR scoring chart. Neonate-related factors (sex, and birth weight) were controlled for in the logistic regression analysis as shown on Table 4.

## Discussion

### Effective use of the APGAR score

We estimated a sensitivity of 71% and a specificity of 89%. Our specificity was more comparable to that of a study prospective cohort study conducted among 464 neonates admitted to a

tertiary hospital in Iran at 81% [11]. On the other hand, we had a lower sensitivity of 71% compared to the 81% in Iran [11].

We postulate that, our low sensitivity can be explained by a form of ascertainment bias, which is also in line with findings from a review on the APGAR score by the score's founder, Dr. Virginia Apgar. She recommended in this review that, the person carrying out the delivery shouldn't be the one assigning the APGAR score as they could have a vested interest in the neonatal outcome, and also have divided attention in view of the need for continual maternal management [9,29]. The practice for most deliveries in MTRH except caesarean deliveries is that the person carrying out the delivery also assigns the APGAR score to the neonate. This could have therefore resulted in incorrectly high APGAR scores, and true asphyxia being reported (per APGAR score) as no asphyxia, translating to low sensitivity. On the flip side, the scores observed in our study were better at determining babies who did not have birth asphyxia.

A multicenter study that assessed accuracy of the APGAR score and umbilical cord blood parameters in diagnosing birth asphyxia, reported an APGAR score sensitivity of 99% and a specificity of 41% [20]. These findings were dissimilar to ours in that, our sensitivity was lower at 71%, and our specificity was higher than theirs at 89%. The difference could be explained by the fact that in the multicenter study, APGAR scores in the first minute were used; while in our study, we used APGAR scores in the fifth minute in our operational definition for birth asphyxia. The use of fifth-minute APGAR scores in the current study was adopted from the combined consensus of the American College of Obstetricians and Gynecologists/American Academy of Pediatrics (ACOG/AAP).

We observed a positive predictive value of 0.62 and a negative predictive value of 0.92. Park and associates in their study reported a positive predictive value of 19.4 and a negative predictive value of 96.2 [19]. As clearly shown, our negative predictive values are comparable. On the other hand, they had a much lower positive predictive value when compared with ours. This difference could be explained by two things. Firstly, they applied the APGAR score in their study as a predictive tool for neonatal mortality, while we used it for diagnosis of asphyxia in our study. Secondly, they included extremely low birth weight babies in their study, and we did not.

## Healthcare provider factors affecting effective use of APGAR score

In the current study, mode of delivery, need for resuscitation and lack of access to an APGAR scoring chart were significantly associated with the effective use of the APGAR score.

Mode of delivery was significantly associated with incorrect APGAR scores whereby, babies born via instrumental deliveries were 8 times more likely to have incorrect scores when compared with neonates born via spontaneous vaginal delivery. Additionally, neonates born via caesarean deliveries were slightly less likely to be assigned incorrect scores with an odds ratio of 0.13. Our findings can be attributed to the concept of normalcy by which, deliveries requiring intervention are already viewed as a deviation from the norm—spontaneous vaginal deliveries, and thus have a beforehand expectation of poor neonatal outcome. Therefore, it is possible that this could have negatively influenced how effectively the APGAR score was used in deliveries requiring intervention in our study. This was however more applicable to instrumental deliveries than caesarean births in our study and we think it was so because, we included babies born via elective caesarean sections with healthy outcomes in which case, the assigning of APGAR scores is forthright. Our findings agreed to a previous study which reported that neonates born via normal vaginal deliveries were more likely to be better scored than their counterparts born through other methods of delivery [22].

Babies who underwent resuscitation in our study were 23.83 times more likely to be assigned incorrect APGAR scores. We presuppose that, this finding could be attributed to the conventional APGAR score itself, as it does not take into account neonatal resuscitation. Additionally, given that the person resuscitating is also the one who scores the neonate, there may have been an element of ascertainment bias whereby, some scores could have been assigned incorrectly to evade responsibility of negative neonatal outcomes. Our findings mirror those reported in another developing country where a significant variation was noted in the APGAR scores assigned to resuscitated versus non-resuscitated babies [27]; and those from a developed country, where the authors [30] observed great variability in the respiratory component of the APGAR score among healthcare providers in the context of resuscitation. Furthermore, in Richmond-Virginia [21], subjective APGAR scoring by healthcare providers in the setting of neonatal resuscitation was detected, leading the authors to recommend the use of other scores such as the Expanded APGAR score or Neonatal Resuscitation and Adaptive score to replace the conventional APGAR score in resuscitated babies. We also highlight this need for alternative scoring methods for resuscitated babies.

Healthcare practitioners without access to an APGAR scoring chart were significantly more likely to incorrectly classify asphyxia compared to healthcare practitioners with access to an APGAR scoring chart. We postulate that, having access to an APGAR scoring chart could have resulted in better utilization of the score in terms of more objective application of the score parameters, as opposed to attempting to apply the score off by heart without the score chart. Our findings were identical to those reported in Nigeria [31], a study carried out in a similar low-resource setting like the current study. Additionally, recommendations have been made regarding a review of the guidelines of the use of the APGAR score among healthcare providers [32].

## Conclusion and recommendations

This study reports low sensitivity and positive predictive values of assigned APGAR scores at 71% and 62% respectively among newborns attending a tertiary hospital in Western Kenya. Newborns who were resuscitated, born by instrumental deliveries, and assigned APGAR scores by healthcare providers without access to the APGAR scoring chart were significantly more likely to get incorrect APGAR scores. With the foregoing, there is need to adopt the gold-standard of birth asphyxia diagnosis into the hospital-based neonatal management guidelines at the tertiary hospital and other resource constrained healthcare settings. In addition, improved APGAR scoring is attainable by augmenting availability of APGAR scoring charts, continuous medical education programs on the score, and on neonatal resuscitation. Additional studies should be conducted to explore the applicability of other neonatal assessment scores for resuscitated neonates.

## Supporting information

**S1 File. Supplementary Material 1: Data Collection Form (Maternal and Neonatal).**
(DOCX)

**S2 File. Supplementary Material 2: Data Collection Form (Healthcare Workers).**
(DOCX)

**S3 File. Supplementary Material 3: Standard Operating Procedure for Umbilical Cord Blood Collection.**
(DOCX)

**S4 File. Supplementary Material 4: Dataset.**
(XLSX)

## Author Contributions

**Conceptualization:** Albertine Enjema Njie, Winstone Mokaya Nyandiko, Phinehas Ademi Ahoya, Jude Suh Moutchia.

**Data curation:** Albertine Enjema Njie, Phinehas Ademi Ahoya, Jude Suh Moutchia.

**Formal analysis:** Albertine Enjema Njie, Winstone Mokaya Nyandiko, Phinehas Ademi Ahoya.

**Investigation:** Albertine Enjema Njie, Winstone Mokaya Nyandiko, Phinehas Ademi Ahoya.

**Methodology:** Albertine Enjema Njie, Winstone Mokaya Nyandiko, Jude Suh Moutchia.

**Project administration:** Albertine Enjema Njie.

**Supervision:** Winstone Mokaya Nyandiko, Phinehas Ademi Ahoya.

**Validation:** Albertine Enjema Njie, Jude Suh Moutchia.

**Visualization:** Jude Suh Moutchia.

**Writing – original draft:** Albertine Enjema Njie, Winstone Mokaya Nyandiko, Phinehas Ademi Ahoya, Jude Suh Moutchia.

**Writing – review & editing:** Winstone Mokaya Nyandiko, Phinehas Ademi Ahoya, Jude Suh Moutchia.

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
