## [Decision Letter · Decision Letter 0]

21 Nov 2022

PONE-D-22-21860Assessment of effective use of APGAR score in comparison to the gold standard in diagnosis of birth asphyxia at Moi Teaching and Referral Hospital, Kenya.PLOS ONE

Dear Dr. Njie,

Thank you for submitting your manuscript to PLOS ONE. After careful consideration, we feel that it has merit but does not fully meet PLOS ONE’s publication criteria as it currently stands. Therefore, we invite you to submit a revised version of the manuscript that addresses the points raised during the review process.

ACADEMIC EDITOR:We suggest that authors make their anonymous data supporting the work to be publicly available unless there is any restriction which should be explicitly stated.

We look forward to receiving your revised manuscript.

Kind regards,

Gbenga Olorunfemi, MBBS,MSC,FMCOG,FWASC

Academic Editor

PLOS ONE

Journal Requirements:

3. You indicated that you had ethical approval for your study. In your Methods section, please ensure you have also stated whether you obtained consent from parents or guardians of the minors included in the study or whether the research ethics committee or IRB specifically waived the need for their consent.

Reviewers' comments:

Reviewer's Responses to Questions

**Comments to the Author**

1. Is the manuscript technically sound, and do the data support the conclusions?

Reviewer #1: Yes

Reviewer #2: Yes

2. Has the statistical analysis been performed appropriately and rigorously? 

Reviewer #1: Yes

Reviewer #2: Yes

3. Have the authors made all data underlying the findings in their manuscript fully available?

Reviewer #1: Yes

Reviewer #2: Yes

4. Is the manuscript presented in an intelligible fashion and written in standard English?

Reviewer #1: Yes

Reviewer #2: No

5. Review Comments to the Author

Reviewer #1: The article is technically and scientifically sound. The only suggested typo identified was the concomitant use of "<" "less than". The article was well written. The style of writing, the language and the soundness of the article is commendable.

Reviewer #2: General comment: Consider employing Professional English Language editing services for this manuscript.

Title: Consider modifying the title to ' A comparative analysis of APGAR score and the gold standard in the diagnosis of birth asphyxia at a tertiary health facility in Kenya.

Short title: Expunge repeated phrase 'compared to'

Line 66-69: A synonymous phrase for neonatality mortality like 'neonatal death' can be used.

Line 84- Major cause or contributor

Line 87 - Consider using a linking word for clarity.

Methods: How were the participants recruited? How did the authors arrive at the total number of participants to be used (sample size)? Consider using a flow chart for these aspects.

Results- 64% is not majority, it's about two-thirds

line 217,234-240 should be rephrased

There should be a consistent use of decimal points in the tables.

Discussion: lines 216-219 have previously been discussed in the abstract and introduction.

Line 224-225- This should be in the conclusion and recommendation

References - Most references used are over 10 years old. Aren't there are more recent articles on the subject matter?

6. PLOS authors have the option to publish the peer review history of their article (what does this mean?). If published, this will include your full peer review and any attached files.

Reviewer #1: **Yes: **M.A.N. ADEBOYE

Reviewer #2: No

---

## [Author Response · Author response to Decision Letter 0]

24 Jan 2023

I have cited Tables 2 and 4 in the text.

---

## [Decision Letter · Decision Letter 1]

28 Mar 2023

PONE-D-22-21860R1A comparative analysis of APGAR score and the gold standard in the diagnosis of birth asphyxia at a tertiary health facility in Kenya.PLOS ONE

Dear Dr. Njie,

Thank you for submitting your manuscript to PLOS ONE. After careful consideration, we feel that it has merit but does not fully meet PLOS ONE’s publication criteria as it currently stands. Therefore, we invite you to submit a revised version of the manuscript that addresses the points raised during the review process.

We look forward to receiving your revised manuscript.

Kind regards,

Gbenga Olorunfemi, MBBS,MSC,FMCOG,FWASC

Academic Editor

PLOS ONE

Journal Requirements:

Reviewers' comments:

Reviewer's Responses to Questions

**Comments to the Author**

1. If the authors have adequately addressed your comments raised in a previous round of review and you feel that this manuscript is now acceptable for publication, you may indicate that here to bypass the “Comments to the Author” section, enter your conflict of interest statement in the “Confidential to Editor” section, and submit your "Accept" recommendation.

Reviewer #2: All comments have been addressed

2. Is the manuscript technically sound, and do the data support the conclusions?

Reviewer #2: Yes

3. Has the statistical analysis been performed appropriately and rigorously? 

Reviewer #2: Yes

4. Have the authors made all data underlying the findings in their manuscript fully available?

Reviewer #2: Yes

5. Is the manuscript presented in an intelligible fashion and written in standard English?

Reviewer #2: Yes

6. Review Comments to the Author

Reviewer #2: Dear author, consider the following minor corrections.

Line 133: Methods: Rephrase to interviewer-administered, not interviewer-filled

Line 165: Result: 51% is approximately half, not majority. Consider rephrasing that.

7. PLOS authors have the option to publish the peer review history of their article (what does this mean?). If published, this will include your full peer review and any attached files.

Reviewer #2: No

---

## [Author Response · Author response to Decision Letter 1]

29 Apr 2023

I would like to submit corrections for my manuscript entitled "Assessment of effective use of APGAR score in comparison to the gold standard in diagnosis of birth asphyxia at Moi Teaching and Referral Hospital, Kenya." 

Reviewer #2: Dear author, consider the following minor corrections.

Comment 1: Line 133: Methods: Rephrase to interviewer-administered, not interviewer-filled.

Response to Comments 1: The methods section (now Line 131) has been rephrased to “Interviewer-administered semi-structured questionnaires”.

Comment 2: Line 165: Result: 51% is approximately half, not majority. Consider rephrasing that.

The results section (now Line 163) has been rephrased to “slightly more than half were male, 52 (51%)”. 

Yours faithfully,

Corresponding Author,

Dr. Njie Albertine.

---

## [Editor Report · Decision Letter 2]

3 May 2023

A comparative analysis of APGAR score and the gold standard in the diagnosis of birth asphyxia at a tertiary health facility in Kenya.

PONE-D-22-21860R2

Dear Dr. Njie,

We’re pleased to inform you that your manuscript has been judged scientifically suitable for publication and will be formally accepted for publication once it meets all outstanding technical requirements.

Kind regards,

Gbenga Olorunfemi, MBBS,MSC,FMCOG,FWASC

Academic Editor

PLOS ONE